# Susceptibility of the Gypsy Moth *Lymantria dispar* (Lepidoptera: Erebidae) to *Nosema pyrausta* (Microsporidia: Nosematidae)

**DOI:** 10.3390/insects12050447

**Published:** 2021-05-14

**Authors:** Anastasia G. Kononchuk, Vyacheslav V. Martemyanov, Anastasia N. Ignatieva, Irina A. Belousova, Maki N. Inoue, Yuri S. Tokarev

**Affiliations:** 1All-Russian Institute of Plant Protection, Podbelskogo 3, Pushkin, 196608 St. Petersburg, Russia; akononchuk@vizr.spb.ru (A.G.K.); edino4estvo@mail.ru (A.N.I.); 2Institute of Systematics and Ecology of Animals SB RAS, Frunze 11, 630091 Novosibirsk, Russia; martemyanov79@yahoo.com (V.V.M.); belousova_i@yahoo.com (I.A.B.); 3Reshetnev Siberian State University of Science and Technology, Krasnoyarskiy Rabochiy av. 31, 660037 Krasnoyarsk, Russia; 4Institute of Biology, Irkutsk State University, Karl Marx Street 1, 664003 Irkutsk, Russia; 5Department of Agriculture, Tokyo University of Agriculture and Technology, Fuchu, 3 Chome-8-1 Harumicho, Tokyo 183-8538, Japan; makimaki@cc.tuat.ac.jp

**Keywords:** microsporidia, host range, gypsy moth, parasite–host interactions, substitute host

## Abstract

**Simple Summary:**

Microsporidia are widespread insect pathogens and a single species may infect different hosts. *Nosema pyrausta* from the corn borer was tested against the gypsy moth. Thirty days after larvae were fed with spores, infection was established in the salivary glands and fat body of pupae and prepupae. Up to 10% of insects became infected. The gypsy moth can be referred to as a resistant host of *N. pyrausta*.

**Abstract:**

The gypsy moth, *Lymantria dispar*, is a notorious forest defoliator, and various pathogens are known to act as natural regulators of its population density. As a widespread herbivore with a broad range of inhabited areas and host plants, it is potentially exposed to parasitic microorganisms from other insect hosts. In the present paper, we determined the susceptibility of gypsy moth larvae to the microsporidium *Nosema pyrausta* from the European corn borer, *Ostrinia nubilalis*. Gypsy moth samples from two localities of Western Siberia were used. *N. pyrausta* developed infections in the salivary gland and adipose tissue of gypsy moth prepupae and pupae, forming spore masses after 30 days of alimentary exposure to the second instar larvae. Among the experimental groups, the infection levels ranged from 0 to 9.5%. Effects of a covert baculovirus infection, phenylthiourea pretreatment and feeding insects on an artificial diet versus natural foliage were not significant in terms of microsporidia prevalence levels. Thus, *L. dispar* showed a low level of susceptibility to a non-specific microsporidium. It can be referred to as a resistant model host and not an appropriate substitute host for laboratory propagation of the microsporidium.

## 1. Introduction

The gypsy moth, *Lymantria dispar* L. (Lepidoptera: Erebidae), is one of the most widespread forest defoliators, which induces outbreaks in most parts of its range in the Holarctic and is the target for pest control in many European, Asian and North American countries [1].

Microsporidia constitute a group of obligate intracellular animal parasites. They are most abundant in insects, and gypsy moth populations harbor several species of these pathogens [2]. Microsporidia have been reported with high prevalence in European populations of gypsy moth and are considered important to the mortality of their hosts [3]. Attempts were made to control gypsy moth populations with microsporidia both in Europe and North America [2,4].

Biotic factors are highly important for insect density dynamics [5]. In European gypsy moth populations, the natural prevalence levels of microsporidia were noted to increase during the period preceding a gypsy moth outbreak and persist at low levels between the outbreaks. Meanwhile, viruses were critical for the termination of a microsporidian outbreak [2].

The host range of some microsporidia is relatively wide and dissemination of a particular species between hosts from different insect orders has been reported [6,7,8,9]. In some cases, their effect can be more detrimental in new hosts than in the typical (adapted) ones [10].

Various microsporidia species were detected in *L. dispar* and some of them were assayed for infectivity in other lepidopteran hosts belonging to different genera and families [11,12,13]. However, studies of susceptibility of the gypsy moth to microsporidia from other hosts (the non-specific parasites) are limited [14]. Some microsporidia species are used for pest control and this means that non-target insects could be challenged by those microsporidia. Moreover, occasional infections due to natural contamination by non-specific microsporidia could also occur in nature. Thus, the aspects of interaction of non-specific microsporidia with non-target hosts remain unexplored. Clarification of these issues are important for understanding patterns of the parasite–host interactions, as well as for biocontrol issues [10].

In the present study, the interaction of *Nosema pyrausta* (Paillot) with *L. dispar* was chosen to be investigated. It is a widespread pathogen of corn borers of the genus *Ostrinia* and is able to infect other Lepidoptera [15], as well as hymenopteran parasitoids [16]. This microsporidium represents, therefore, a non-specific parasite species with an ability to host shift. Testing its infectivity in non-target insect hosts contributes to our understanding of microsporidia host ranges and may have broader applications, such as pathogen circulation in natural ecosystems, substitute host selection for biotechnological production, etc.

The following two questions were addressed in the current study: (i) whether the gypsy moth is susceptible to *N. pyrausta* and (ii) whether infection with a non-specific species of microsporidia may be augmented by environmental factors causing immunosuppression in the gypsy moth.

## 2. Materials and Methods

### 2.1. Insects and Pathogen Cultures

The gypsy moth, *L. dispar*, from Western Siberia, Russia, was used as the model host. The progeny of approximately 200 females were collected as egg masses from two localities in the Novosibirsk region, corresponding to the two local populations. According to our recent mtDNA data, insects from both studied localities belong to same metapopulation [17]. The locality #1 originated from Bazovo, Novosibirsk Region, Russia (54.56° N, 81.21° E) and was at the phase of the rise in population density. Conversely, the insects from locality #2 (Bespyatyy, Novosibirsk Region, Russia 54.14° N, 79.71° E) were at the phase of collapse in population density. Approximately 100 egg masses were collected from locality #1 and another sample of 100 egg masses was obtained from locality #2. These two insect cohorts were additionally characterized for the presence of covert *Lymantria dispar* multiple nucleopolichedrovirus (LdMNPV), as it is known to differ significantly depending on the phase of the host population cycle [18]. The collected egg masses were stored in a refrigerator at +4 °C for winter diapause from September to April [19]. Before the beginning of the experiments (at the end of diapause), the egg masses were cleaned of setae, homogeneously mixed as described by Martemyanov et al. [20] and then surface sterilized with sodium hypochlorite according to the method of Doane (1969). The egg stocks were sampled to estimate the covert LdMNPV prevalence levels and for bioassays as described further.

For the diapause interruption, the egg masses were placed into a thermal incubator set at 28 °C for 3 days prior to mass larval hatching according to the routine procedure for this species [19,20]. Hatched larvae were maintained in 20 L plastic containers (100 individuals per container) and fed with branches of silver birch, *Betula pendula* Roth, its primary host plant for the studied population. The tree leaves were preliminarily washed with sterile water to decrease the risk of contaminating the larvae with wild LdMNPV.

*Nosema pyrausta* [21] was propagated in a temporary laboratory culture of the European corn borer, *Ostrinia nubilalis* Hbn, which is the typical host of this parasite. The larvae were maintained on an artificial diet [22] and infected with *N. pyrausta* spores according to Tokarev et al. [15]. To isolate the parasite spores, infected insects were dissected, adipose tissue was homogenized using plastic pestles and centrifuge tubes in distilled water and spun at 500–1000 g for 5–10 min. Debris was removed using pipette tips, and spore pellets were stored in water for 2–3 months at +8 °C prior to the experiments. Spore concentrations were scored using Neubauer’s hemocytometer (Merck, Darmstadt, Germany).

Infectivity of the microsporidia spore batches used to infect the gypsy moth larvae (see below) was confirmed by standard bioassays using *O. nubilalis* [23], showing a 90–100% infection level in the test insect groups as opposed to the control groups showing no infection.

### 2.2. Estimation of Prevalence Levels of Covert LdMNPV Infection in L. dispar Populations

As coexisting infections may alter the interactions of the target parasite with its insect host, we tested the host samples for prevalence of covert LdMNPV, which may persist within the *L. dispar* populations [18]. A pooled sample of 10 randomly chosen eggs was taken from egg mass mixtures of each studied locality (see above) to representatively determine the presence of covert LdMNPV by PCR. Ten pooled samples per locality (i.e., in total 100 eggs per population) were studied. For the positive samples, as many as 30 additional randomly picked eggs were analyzed individually to verify the percentage of the stock infected with LdMNPV. Total DNA was extracted using the phenol-chloroform method [24], with some modifications. Egg masses were mechanically homogenized with a pestle in a lysis solution containing guanidine thiocyanate (Amresco, Solon, OH, USA). For detection with higher sensitivity, qPCR was used for the qualitative analysis of the virus, as we described in detail previously [25] using the CFX-96 Touch™ Real-Time PCR Detection System (Bio-Rad Laboratories, Hercules, CA, USA).

In populations for which LdMNPV presence was confirmed, late instar larvae were additionally assayed for covert infection to examine persistence of the baculovirus during insect development. This was done using the same technique applied to the DNA extracts from inner tissues (fat body), as described below.

### 2.3. Microsporidia Infection Assay

To test the susceptibility of *L. dispar* to infection with *N. pyrausta*, a series of artificial infection experiments were performed. To increase chances of infection with this non-specific parasite, young larvae (second instar) were used, which are more vulnerable to entomopathogens [26]. Insects were sampled from two different localities at two contrast population phases (rise vs. decline) suggesting differential susceptibility to infections [27]. To induce depression of immune status, one half of both insect samples was pretreated with phenylthiourea (PTU; Merck, Darmstadt, Germany) which causes specific immunosuppression in insects by inhibition of the prophenoloxidase cascade [28]. Finally, all the experimental variants were split into two equal groups, one fed with natural forage (birch leaves) and another with an artificial diet, which is known to increase host susceptibility to alimentary infections [29].

Briefly, the bioassay scheme was as following (see details below): (1) collection of II instar larvae reared from eggs collected from two localities; (2) 24 h pretreatment with PTU or water as control; (3) overnight starvation of larvae and alimentary treatment with microsporidia spores or water as control; and (4) separation of each experimental group into two subgroups, fed on birch leaves and on an artificial diet, respectively (Figure 1), with four replicates in each variant.

As a result, the following variants were assayed for microsporidia infection:-Naïve insects fed on birch leaves;-Naïve insects fed on an artificial diet;-Immunosuppressed insects fed on birch leaves;-Immunosuppressed insects fed on an artificial diet.

This principal scheme was exploited to test the susceptibility of insects from two localities (#1 and #2) with opposite phases of population cycle to the microsporidium *N. pyrausta*.

To model insect immunosuppression using PTU pretreatment, birch leaves were sprayed with an aqueous solution of 0.1% PTU (or distilled water as control) and fed to the newly molted second instar larvae. After 24 h, the forage was removed and the larvae were starved overnight. For microsporidia treatment, dosed spore suspensions, adjusted to 16 × 10^7^ spores, were carefully applied to both sides of a limited number of birch leaves, air dried for 10–15 min and exposed to the experimental groups of larvae (Figure 2A), either pretreated or not pretreated with PTU. Each treatment group consisted of 80 larvae, and the mean infection dose was 2 × 10^6^ spores/larvae. The control larvae were fed with leaves not treated with microsporidia spores. After complete consumption of contaminated leaves (Figure 2B), the larvae were transferred to 0.5-mL plastic containers (10 larvae per container) and supplied with birch leaves or an artificial diet [20,30] according to the experimental scheme above.

### 2.4. Detection of Pathogens through Infection Assays

For light microscopy detection of microsporidia, smears were prepared from inner tissues (fat body, salivary glands and midgut epithelium) of newly perished fresh larvae or from whole-body homogenates of macerated/dried cadavers (where inner tissues could not be sampled). Smears were examined at a minimum of 30 fields of view using a bright field Axio 10 Imager M1 microscope (Carl Zeiss, Oberkochen, Germany) at 400–1000× magnification. Samples that were positive for microsporidia spores were subjected to routine genomic DNA extraction [31]. PCR was run with a standard protocol using DreamTaq DNA polymerase (Thermo Fisher Scientific, Waltham, MA, USA) as a ready-to-use mixture (https://www.thermofisher.com/order/catalog/product/EP0701, accessed on 12 May 2021). The partial small subunit ribosomal RNA (SSU rRNA) gene of the microsporidia was amplified using 18f and 1047r primers [32]. The PCR products were directly sequenced using an ABI Prism Genetic Analyzer 3500 (Applied Biosystems, Waltham, MA, USA). Alternatively, the *N. pyrausta-*specific NpyrHKfor1:NbHKrev1 primers were used for direct PCR detection of the pathogen [33].

### 2.5. Statistical Analysis

Statistical analysis was performed to elucidate the effect of experimental conditions on survival and microsporidia prevalence levels (where applicable) using a generalized linear model (GLM) with binary distributions and the logit link function in Statistica 7.0 (StatSoft, Inc., Tulsa, OK, USA). Locality, PTU pretreatment, *N. pyrausta* treatment and forage type were set as categorical predictors (factors), and mortality was the binary dependent variable.

## 3. Results

### 3.1. Prevalence of Covert LdMNPV in the Lymantria dispar Populations

In Western Siberia, LdMNPV is widespread in gypsy moth populations in the form of a covert infection [18,30,34]. It was therefore necessary to consider the presence of this pathogen as it might have influenced the interactions of the microsporidia and the gypsy moth in the present study.

Insects from locality #1 were free from covert LdMNPV. No positive response was detected when 10 pooled samples (10 eggs in each sample) were studied by qPCR. In contrast, the insect sample from locality #2 was totally infected by covert LdMNPV, as 100% of egg samples (number of examined specimens = 130) and fully grown larvae (*n* = 10) were found positive for infection, according to the results of the qPCR analysis.

### 3.2. Insect Survival Estimates

The second instar gypsy moth larvae from locality #1 fed with *Nosema pyrausta* spores and maintained on birch leaves successfully completed larval development and proceeded to pupation at the level of 72.5 ± 7.06% (mean ± SE, number of assayed insects *n* = 40) when not pretreated with PTU (naïve). Similarly, 70 ± 7.26% (*n* = 40) of PTU-pretreated insects pupated. When an artificial diet was used instead of fresh birch leaves, the pupation level estimates reached only 55.5 ± 7.86% (*n* = 40) and 52.5 ± 7.90% (*n* = 40), respectively. Survival levels in *N. pyrausta-*treated groups were about the same as in the respective control groups (Figure 3).

The survival estimates for locality #2 were lower. When fed with birch leaves, naïve and PTU-pretreated larvae showed 55 ± 7.87% (*n* = 40) and 52.5 ± 7.90% (*n* = 40) survival, respectively. When fed with an artificial diet, naïve and PTU-pretreated larvae showed 40 ± 7.75% (*n* = 40) and 35 ± 7.54% (*n* = 40) survival, respectively. Again, these values were similar to those of the respective control groups (Figure 3).

Statistical analysis using a GLM model showed the following. The effects of locality (df = 1, W = 19.413, *p* = 0.000011) and diet type (df = 1, W = 15.486, *p* = 0.000083) on survival levels were statistically significant at the 0.01 significance level. Conversely, the effects of PTU pretreatment (df = 1, W = 0.137, *p* = 0.711) and microsporidia (df = 1, W = 1.07633, *p* = 0.299) treatment were not significant.

### 3.3. Microsporidia Prevalence Levels

*N. pyrausta* infection was revealed in several pupae and prepupae across the experimental groups treated with parasite spores. The parasite spore masses were located in salivary gland and/or adipose tissue cells (Figure 2C,D). In the sample from locality #1, 0% (number of examined insects *n* = 29) and 3.6 ± 3.52% (*n* = 28) of the insects fed on birch leaves were infected in the naïve and PTU-pretreated groups, respectively, and 4.3 ± 4.23% (*n* = 23) and 9.5 ± 6.40% (*n* = 21) of the insects fed on artificial diet were infected in the naïve and PTU-pretreated groups, respectively. In locality #2, the microsporidia prevalence levels were 4.5 ± 4.42% (*n* = 22) and 9.5 ± 6.40% (*n* = 21) in naïve and PTU-pretreated larvae, respectively, while insects fed on an artificial diet showed respective values of 6.3 ± 6.05% (*n* = 16) and 7.1 ± 6.88% (*n* = 14). The effects of locality (df = 1, W = 0.248, *p* = 0.619), diet type (df = 1, W = 0.248, *p* = 0.619) and PTU pretreatment (df = 1, W = 0.420, *p* = 0.517) were not statistically significant (Figure 4).

## 4. Discussion

Microsporidia infections in the gypsy moth are frequent in European and North American populations and have been extensively studied under natural and experimental conditions [2,4,13,35]. Although microsporidia infections were not revealed during surveys of gypsy moth populations in the Novosibirsk region [36], it is known that certain species of microsporidia are able to infect non-specific hosts. For example, *Nosema bombycis* and *Vairimorpha necatrix* were isolated from representatives of several families of Lepidoptera [37].

The present study was therefore aimed at challenging gypsy moth larvae with a microsporidium species that typically develops in another lepidopteran host to test insect susceptibility to this pathogen. Because the prophenoloxidase system is involved in insect immunity to pathogens, including microsporidia [38] but not viruses [28], PTU in vivo pretreatment was used at a concentration that significantly inhibits PO activity in *L. dispar* hemolymph without causing an antifeedant effect in the larvae [28].

*N. pyrausta* showed a principal ability to parasitize *L. dispar* but was only slightly infective. Approaches based on immunosuppression in this study, such as PTU pretreatment, tended to increase the prevalence level of the non-specific microsporidium, though the limited sample size did not allow us to reveal a significant difference as compared to naïve insects challenged with *N. pyrausta.* A similar observation was made in a previous study, where the prevalence level of this microsporidium in another substitute host (*G. mellonella*) tended to increase under the influence of either PTU or *Bacillus thuringiensis* pretreatments, but a significant increase (as compared to control) was observed only when both stressors (PTU and *B. thuringiensis*) were combined [15]. Thus, inhibition of insect prophenoloxidase system itself does not cause drastic changes in the infection process, suggesting that insect immunity to microsporidia is more complex.

The insects from the two subpopulations (sampled from the two localities, respectively) showed significantly different survival levels when compared between respective variants of diet types (Figure 3), and this corresponded to their population cycle phase (rising vs. collapsing) accompanied with contrasting levels (0% vs. 100%) of the covert baculovirus infection. Diet type was also found to affect the fitness of gypsy moths from both localities, as survival levels were significantly lower in the artificial diet-fed larvae as compared to the birch-fed ones (Figure 3). Additionally, though survival levels were about the same in microsporidia-challenged and control insects, the response to microsporidia in terms of the parasite prevalence levels tended to be different in the two gypsy moth subsamples depending upon the diet type. Unfortunately, the sample size was not great enough to support the observations statistically, but we can speculate that insects in the rising phase are less susceptible to infection when fed with fresh birch leaves (resulting in microsporidia prevalence levels of 0–3.6%) as compared to artificial diet variants (4.3–9.5%). This suggestion fairly corresponds to the fact that feeding with an artificial diet suppresses insect resistance to alimentary infections, in particular, by decreasing the peritrophic membrane thickness [29]. Meanwhile, insects at the collapse phase display decreased fitness and their resistance to infections is lower in both cases, suggesting that diet type is not important in this case and optimal forage quality (exemplified by fresh birch leaves) does not facilitate resistance to the disease.

## 5. Conclusions

The following conclusions may be drawn from the data obtained through the present study. First, the microsporidium *N. pyrausta* is only slightly infective to the second instar gypsy moth larvae and does not have a detrimental effect on insect survival. Second, the immunosuppressive state of the gypsy moth larvae, caused by various environmental factors, does not significantly affect infection levels by *N. pyrausta*. Third, diet type has a notable influence on insect fitness and in certain cases tends to change gypsy moth susceptibility to microsporidia. Finally, the gypsy moth can be considered as a resistant model host and not an appropriate substitute host for laboratory rearing of *N. pyrausta*.

## Figures and Tables

**Figure 1 insects-12-00447-f001:**
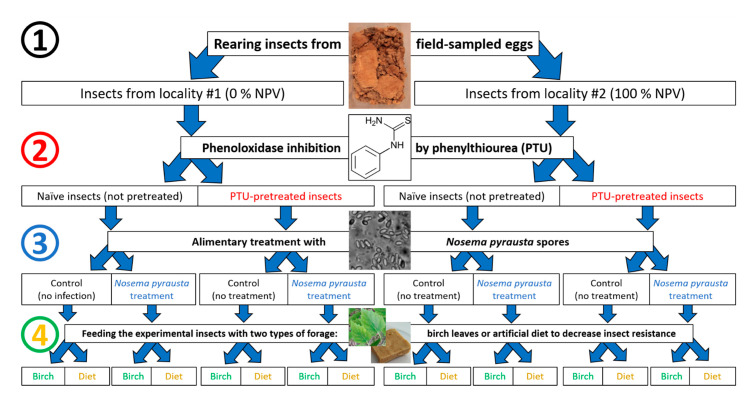
Experimental scheme showing the four stages of insect manipulation, corresponding to the four factors examined: (**1**) sampling from two localities, (**2**) pretreatment with phenylthiourea (PTU) or no pretreatment (naïve), (**3**) treatment with microsporidia or no treatment (control) and (**4**) feeding either with fresh birch leaves or an artificial diet.

**Figure 2 insects-12-00447-f002:**
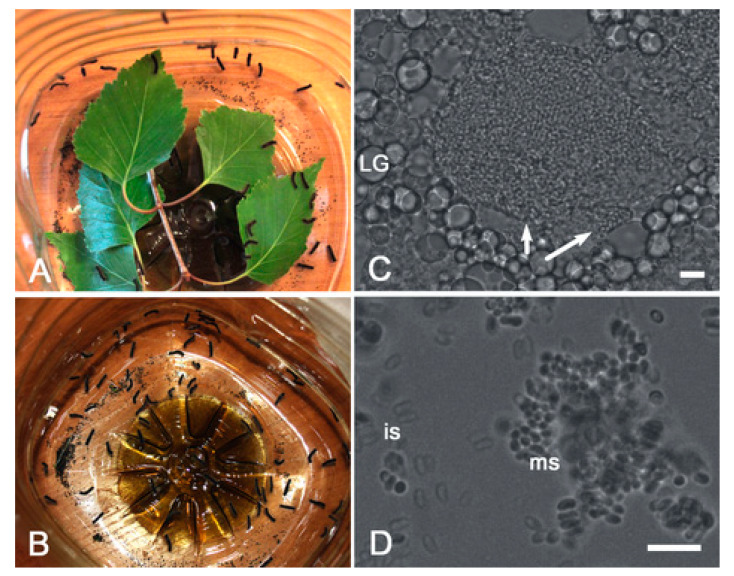
Infection of *Lymantria dispar* larvae with microsporidia. (**A**,**B**) A sample group of larvae before (**A**) and after (**B**) the contaminated birch leaves were consumed. (**C**,**D**) Bright field microscopy showing parasite spore-loaded infected cells (arrows) surrounded with lipid granules (LG) of uninfected cells, as well as immature (is) and mature spores (ms) on a smear. Scale bar = 10 micrometers.

**Figure 3 insects-12-00447-f003:**
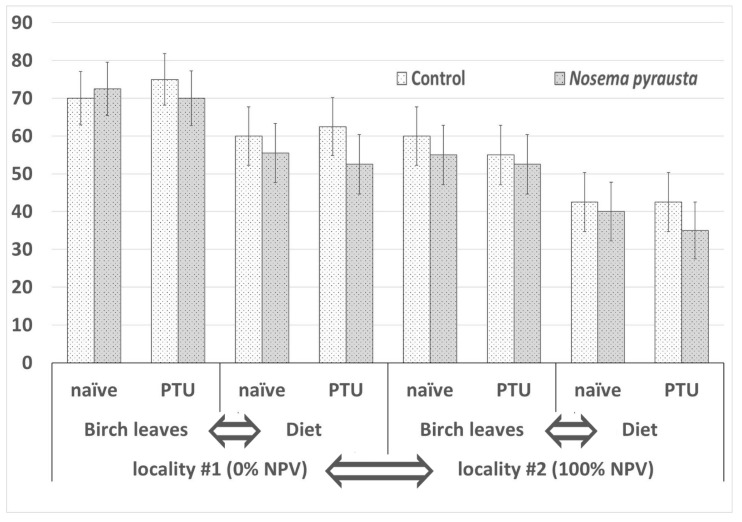
Survival levels in gypsy moth larvae 30 days after challenging with *Nosema pyrausta* spores. A generalized linear model was applied to test effects of four factors according to the experimental scheme shown in Figure 1. Factors causing statistically significant differences between the respective variants (*p* < 0.01) are indicated by arrows.

**Figure 4 insects-12-00447-f004:**
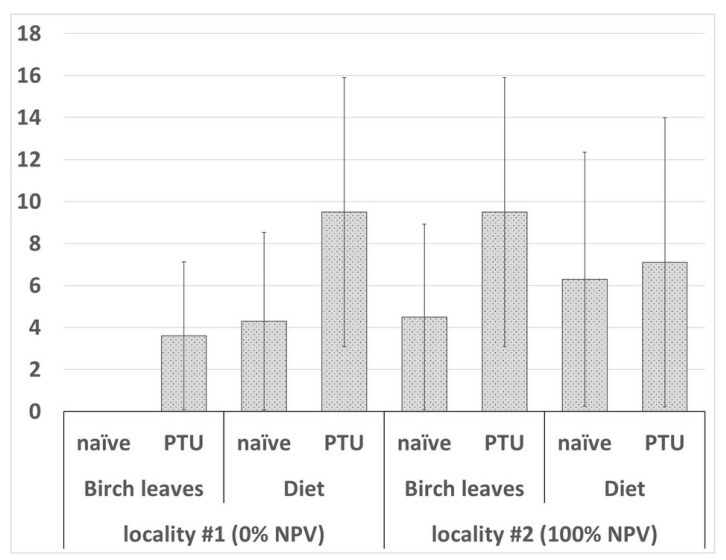
Microsporidia infection level in gypsy moth larvae 30 days after challenging with *Nosema pyrausta* spores. Statistical approach is as described in Figure 3. No statistically significant differences between the variants were found.

## Data Availability

The data presented in this study are available on request from the corresponding author. The public data sharing was not approved by all organizations where authors are affiliated.

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
