# Peer review of "Susceptibility of the Gypsy Moth Lymantria dispar (Lepidoptera: Erebidae) to Nosema pyrausta (Microsporidia: Nosematidae)"

_insects, 2021, doi:10.3390/insects12050447_

Round 1

Reviewer 1 Report

The paper is written well and well executed. The methods and results are clear. I believe that the paper should be excepted with minor changes

increases N. pyrausta infection levels only insignificantly.

Might be:

does not significantly affect infection levels by N. pyrausta

At the same time the paper basically says that Nosema pyrausta does not infect Lymantria dispar. The authors should have their eyes on a shorter paper in the future. Since the authors have a good laboratory set up for testing microsporidia on this important forest defoliator, I would encourage them to test other species of Microsporidia and present the results in a shorter table format. They also might want to keep a laboratory culture of Lymantria dispar for future use if the Gypsy moth is a problem in Russia.

Simple summary-

if statistics doesn't show a difference then leave out the sentence with “under certain conditions”

Introduction -

preceding the host outbreak might be “preceding a gypsy moth outbreak”

Line 112 take out “remarkable”

Author Response

RESPONSE #1

We are thankful to the Reviewer for corrections and suggestions. We want to highlight that Nosema pyrausta is infective to Lymantria dispar but at low prevalence level. These results are compared to its performance in other lepidopteran hosts – both specific (Ostrinia) and non-specific (Galleria). Unfortunately, L. dispar is monovoltinous so we can only perform one series of bioassays per year. Nevertheless, we plan to test some new isolates of Microsporidia from several lepidopteran hosts in the near future and present these data in another paper. Then the results obtained herein could be compared to the new dataset.

The phrase “Under certain conditions, the parasite prevalence levels tended to increase, but the differences were not significant” is removed (Line 29)

The phrase “preceding the host outbreak” is replaced with “preceding a gypsy moth outbreak” (Line 41)

The word “remarkable” is removed (Line 112)

The phrase “increases N. pyrausta infection levels only insignificantly” is replaced with “does not significantly affect infection levels by N. pyrausta” (Line 246)

Reviewer 2 Report

The manuscript is in the scope of the journal Insects. The authors present original results about the susceptibility of the gypsy moth, Lymantria dispar   to the microsporidium Nosema pyrausta and conclude that the pest can be considered as a resistant model host and could not be used for laboratory rearing of N. pyrausta.

Mu suggestion is to accept the paper for publication in Insects. I have only few comments and suggestions shown in the attached file.

Author Response

RESPONSE #2

We are thankful to the Reviewer for corrections and questions, which have been thoroughly revised and answered as listed below:

Line 43: corrected as suggested to “of a microsporidian outbreak”

Line 55: corrected as suggested to “In the present study, the interaction of Nosema pyrausta (Paillot) with L. dispar was chosen to be investigated”.

Lines 66-67: added “corresponding to the two local populations” to address the question in Line 103 of whether two sites mean two populations.

Lines 134-137: here we changed “adjusted to 2×106 spores/larva” to “adjusted to 16×107 spores” and added the phrase “Each treatment group consisted of 80 larvae, and the mean infection dose was 2×106 spores/larvae” To explain that this was not an individual dosing (that would be too laborious for such an extensive bioassay) but treatment of a group providing a MEAN infection dose.